# Molecular Mechanisms of Premature Aging in Hemodialysis: The Complex Interplay between Innate and Adaptive Immune Dysfunction

**DOI:** 10.3390/ijms21103422

**Published:** 2020-05-12

**Authors:** Vincenzo Losappio, Rossana Franzin, Barbara Infante, Giulia Godeas, Loreto Gesualdo, Alberto Fersini, Giuseppe Castellano, Giovanni Stallone

**Affiliations:** 1Nephrology, Dialysis and Transplantation Unit, Department of Medical and Surgical Sciences, University of Foggia, 71100 Foggia, Italy; villy79@yahoo.it (V.L.); barbarainf@libero.it (B.I.); giulia.godeas@alice.it (G.G.); giovanni.stallone@unifg.it (G.S.); 2Nephrology, Dialysis and Transplantation Unit, Department of Emergency and Organ Transplantation, University of Bari, 70124 Bari, Italy; rossanafranzin@hotmail.it (R.F.); loreto.gesualdo@uniba.it (L.G.); 3General Surgery Units, Department of Medical and Surgical Sciences, University of Foggia, 71100 Foggia, Italy; alberto.fersini@unifg.it

**Keywords:** premature aging, complement, kidney, hemodialysis

## Abstract

Hemodialysis (HD) patient are known to be susceptible to a wide range of early and long-term complication such as chronic inflammation, infections, malnutrition, and cardiovascular disease that significantly affect the incidence of mortality. A large gap between the number of people with end-stage kidney disease (ESKD) and patients who received kidney transplantation has been identified. Therefore, there is a huge need to explore the underlying pathophysiology of HD complications in order to provide treatment guidelines. The immunological dysregulation, involving both the innate and adaptive response, plays a crucial role during the HD sessions and in chronic, maintenance treatments. Innate immune system mediators include the dysfunction of neutrophils, monocytes, and natural killer (NK) cells with signaling mediated by NOD-like receptor P3 (NLRP3) and Toll-like receptor 4 (TLR4); in addition, there is a significant activation of the complement system that is mediated by dialysis membrane-surfaces. These effectors induce a persistent, systemic, pro-inflammatory, and pro-coagulant milieu that has been described as inflammaging. The adaptive response, the imbalance in the CD4+/CD8+ T cell ratio, and the reduction of Th2 and regulatory T cells, together with an altered interaction with B lymphocyte by CD40/CD40L, have been mainly implicated in immune system dysfunction. Altogether, these observations suggest that intervention targeting the immune system in HD patients could improve morbidity and mortality. The purpose of this review is to expand our understanding on the role of immune dysfunction in both innate and adaptive response in patients undergoing hemodialysis treatment.

## 1. Introduction

End-stage renal disease (ESRD) is an extremely serious condition, recognized as a public health priority and affecting more than 2.6 million people worldwide.

A large part of patients affected by ESRD are dialysis-dependent for the rest of their life and have an increased risk of cardiovascular morbidity and mortality, but also a higher susceptibility to infections and malignancies [1]. The amplified exposure to clinical complications is related to traditionally described risk factors (such as diabetes mellitus, hypertension, and dyslipidemia), but also to non-traditional risk factors such as the persistent, chronic, and systemic inflammation generally described as “dialysis syndrome”. Several pathophysiological mechanisms are involved in the establishment of chronic inflammation and can be divided in exogenous factors, such as dialysis membranes and central venous catheters contamination, and endogenous factors. The latter includes cellular processes, such as the endothelial dysfunction and cellular senescence; microenvironmental factors, such as oxidative stress, hypoxia, fluid overload, and sodium overload; microbial factors, such as immune dysfunction and gut dysbiosis; and, finally, the retention of uremic toxins, such as indoxyl sulphate, advanced glycation end (AGE) products, and calcio-protein particles [1,2]. (Figure 1)

Several pieces of evidence demonstrated that elderly (aged more than 65 years old) hemodialysis (HD) patients showed a higher risk of developing cardiovascular and neoplastic events, and are more susceptible to infections, respond poorly to standard vaccination procedures, and have an increased risk of virus-associated cancer [2,3] compared with younger subjects [4].

A recent demographical evaluation of maintenance dialysis throughout the world, from 1990 to 2010, showed a substantial growth in the utilization of maintenance dialysis in almost all world regions according to changes in population structure, aging, and the worldwide increase in diabetes mellitus and hypertension [5]. According to USRDS, (United States Renal Data System) in 2016, nearly half of incident dialysis patients in the United States had a median age of 64.4 years old [6], with a similar trend in all Western countries [7]. Furthermore, the elderly, those aged more than 65 years old, are the fastest-growing group of incident dialysis patients [8,9].

Nearly all of these elderly patients employ HD as dialysis treatment [10] and their mortality and survival is strongly influenced by comorbidities such as vascular and cardiac disease, whose mechanisms are linked to inflammation and microvascular damage [11], which are related to a progressive accumulation of AGEs, with increased levels of CRP, PTX3, IL-6 [12,13], and FGF-23 [14,15]. These are all large middle uremic toxins difficult to remove with standard dialyzers and responsible for chronic inflammation.

## 2. The Link between Chronic Inflammation and Premature Renal Aging

Chronic, systemic inflammation is strongly integrated with premature aging. Consistently, data from literature are suggesting that the inflammatory milieu associated with the “dialysis syndrome” can lead to a condition defined as “inflammaging”. Therefore, patients affected by ESRD or in RRT (Renal Replacement Therapy) may represent a population of particular interest in order to evaluate the development of accelerated aging phenomena [16].

ESRD is characterized by specific clinical patterns, consistent with the evidence of sarcopenic obesity, reduced aerobic exercise resistance and glucose uptake, impaired cognitive functions, and depression. At the same time, these changes in body composition, impaired energy balance, impaired homeostatic mechanism, and neurodegeneration define the “aging phenotype” according to geriatric guidelines [17].

Particularly, patients affected by ESRD on RRT present a chronic state of inflammation that can be related to exogenous causes depending on the dialysis treatment itself and the use of low-biocompatible materials, the accumulation of uremic toxins such as advanced glycation end products (AGEs) and protein bound uremic toxins (PBUTs) not efficiently removed during the extracorporeal treatment, and calcium and phosphate retention. At the same time, the coexistence of comorbidities such as cardiovascular disease, previous failed allograft, latent infection, chronic inflammatory diseases with tissue factors such as fluid overload, sodium overload, and RRT-induced hypoxia exert a severe intracellular stress, leading to progressive cellular senescence, oxidative stress with mitochondrial deficiency, and endothelial dysfunction [18]. Interestingly, several factors such as complement activation and increased fibrosis of adventitia are involved in the failure of arteriovenous fistula [19,20]. It is also well documented in the literature that ESRD is associated with a specific serological pattern of inflammation, footprint of chronic renal failure, defined by increased levels of C reactive protein (CRP), IL-6, and TNF-α, which can be considered consequences of the accumulation of uremic toxins derived from intestinal microbiota dysbiosis [21]. TNF and IL-6 are also responsible for the induction of catabolic effects through the stimulation of the ubiquitin proteasome complex, the downregulation of anabolic pathways through IGF (insulin-like growth factor) resistance, and the misregulation of the mTOR pathway [22,23,24]. In addition, oxidative stress, which is considered a major contributor to biological aging, is increased in ESRD and reciprocally related to its uremic milieu [25,26]. It is also known that senescence makes cells more prone to the damage evoked by uremic toxins and/or oxidative stress [23].

This inflammation leads to a “premature aging phenotype” characterized, in HD patients, by muscle wasting, vascular calcifications, and cardiovascular hypertrophy, and all related to generalized and increased catabolic processes. Potential mechanisms of inflammaging in HD patients included the genetic susceptibility (i.e., polymorphism for complement factors such as factor H); the oxidative stress caused by dysfunctional mitochondria; activation of NOD-like receptor P3 (NLRP3) and Toll-like receptors (TLRs); the dysregulation of calcium and phosphate metabolism (such as fetuin and Klotho proteins); and, more importantly, the cellular senescence. Persistent DNA damage and epigenetic changes have been identified as the main causes leading to premature kidney cells senescence [18]. Senescence is commonly described as an arrest in cellular proliferation and the inhibition of apoptosis allowed by the continuous upregulation of checkpoint cell cycle inhibitors such as p16^INK4a^ and p21^WAF1^. Interestingly, senescent cells exhibit an inflammatory phenotype termed the senescence-associated secretory phenotype (SASP), characterized by the secretion of an array of proinflammatory cytokines and growth factors (i.e., IL6, IL8, MCP1, PAI-1, CXCL1) as well as proteases (MMP2, MMP3, and collagenase 3) that can act in a paracrine manner on neighboring cells and tissues. In HD settings, cellular senescence has recently been reported to promote osteogenic differentiation of vascular smooth muscle cells (VSMCs), suggesting there may be a direct link between aging and vascular calcification. For years, for dysregulated calcium and phosphate metabolism, the impaired FGF23-Klotho-vitamin D axis signaling has been linked to the occurrence of osteoporosis, vascular calcification, and premature aging. However, traditionally, other confounding factor have been taken in account as traditional premature aging risk factors such as diabetes, dyslipidemia, hypertension, or smoking. On the contrary, Pilar Sanchis et al. [26] revealed that, independently from age, vessels from patients with Chronic Kidney Disease (CKD) exhibited features of premature aging in vivo, including oxidative DNA damage and elevated senescence markers, and these aging signatures persisted on the culture of VSMCs in vitro. In an elegant study, Pilar Sanchis et al. demonstrate that vessels from children with stage 5 CKD with or without HD had increased oxidative DNA damage, as well as p16 and p21 senescence markers. Vascular smooth muscle cells cultured from children on dialysis exhibited persistent DNA damage, impaired DNA damage repair, and accelerated senescence. Under calcifying conditions, vascular smooth muscle cells from children on dialysis showed increased osteogenic differentiation and calcification. These changes correlated with the activation of SASP. (Figure 1)

Therefore, CKD and HD induce an oxidative DNA damage in vessels that promotes a cascade of events from inflammation, to senescence, arterial stiffening, and vascular calcification. These studies highlight the need for new approach in dialysis treatment capable of lowering inflammaging. Recently, promising results are coming from Sepe et al.’s study, which demonstrated that vitamin E-loaded dialyzer was able to counteract the inflammaging and immunosenescence by reducing IDO1 activity and NO formation [27].

Interestingly, the premature aging process also involves the immune system [21]. Such a condition downregulates both innate and adaptive immune responses during chronic disorders such as type II diabetes, cancer, and hemodialysis, accounting for their susceptibility to infections, malignancy, and resistance to vaccination.

This review summarizes recent findings and pivotal mechanisms of innate and adaptive immune dysfunction in patients undergoing hemodialysis treatment. Moreover, we will review new therapeutic strategies capable of counteracting these molecular and cellular alterations.

## 3. Innate Immune Response during Hemodialysis

### Cellular Compartments: Neutrophils and Monocytes

During the hemodialysis treatment, the most important cells responsible for increased immunoreactivity to uremic toxins or chronic exposure to bio-incompatible membranes have been identified in neutrophils and macrophages-monocytes. Together with other important players of innate response such as dendritic cells and natural killer (NK) cells, these cells express a broad repertoire of PRR (pathogen recognition receptor), namely the Toll-like receptors (TLRs) able to recognize a wide range of PAMPs (pathogen associated molecular patterns) or DAMPs (damage associated molecular patterns), in order to orchestrate an immune response via cytokines and chemokine production.

Recent evidence showed that changes in the number of circulating neutrophils and NK and their altered phagocytic activity reflect the impairment of the native immunity observed in patients on dialysis and can be influenced by the membrane’s biocompatibility and dialysis modality. Nevertheless, several studies of neutrophils number and function in HD patients have produced conflicting results [28,29,30].

Neutrophils, the major cellular component of the innate immune system, are central players in the early response to inflammation and infection. During HD, there is an early, transient decrease in the number of polymorphonuclear cells that underwent apoptosis. The resulting neutropenia is associated with an increased risk of infection and higher infection mortality rates in HD patients. In previous studies, Toren M et al. [31] hypothesized that the neutrophils that disappear from the circulation during the development of dialysis-neutropenia were sequestered in the pulmonary capillaries. Lately, Hoenich N. suggested that membrane-activated neutrophils adhere to the endothelial walls of the pulmonary capillaries, the first vascular surface of contact after leaving the dialyzer [32]. Besides the adhesion to endogenous capillaries, the blood flow on the dialysis filters and in particular on the bio-incompatible material can recruit and activate circulating neutrophils and monocytes. After activation, these cells released several pro-inflammatory cytokines (such as IL-6, IL-1, TNFα) as well as activators of the complement system [33]. However, this dysfunctional activation affects their phagocytic activity, which remains strongly decreased during HD independently from dialysis type or membrane [34]. The neutrophils dysfunction has been reviewed elsewhere [35] and has also been correlated to complement alternative pathway activation [36].

In addition to apoptosis and dysfunctional activation, senescent polymorphonuclear neutrophils have been described to increase during HD. Initially, different studies demonstrated a role for CXCL12/CXCR4 in the turnover of senescent polymorphonuclear cells [37]. Senescent polymorphonuclear cells return to bone marrow and undergo apoptosis. CXCL12 coordinates this event interacting with CXCR4, preferentially expressed on senescent polymorphonuclear cells and inducing TNF-related apoptosis-inducing ligand. Interestingly, in HD patients, Zaza G et al. demonstrated a low-CXCL12 expression, suggesting an excessive accumulation of senescent polymorphonuclear cells, potentially underlying the neutrophils dysfunction observed in these patients. As observed for neutrophils, NK cell count is also significantly lower in dialyzed patients in comparison with healthy subjects [34].

## 4. Humoral Compartments: PRR and Cytokines

The activation of the innate immune system occurs through the activation of TLRs. In the TLR family, TLR-4 signaling is initiated by several PAMPs (i.e. LPS, dsDNA, flagellin) and leads to the activation of NF-kB [38], the main regulator of cytokine secretion. From the other side, NF-kB is also upregulated by oxidative stress and other cytokines, such as TNF, suggesting the hypothesis of a self-sustained mechanism of inflammation in ESRD [39]. In 2014, Jurk and colleagues induced chronic inflammation in non-uremic mice by the knockout of the NF-kB subunit 1, enlightening telomere shortening and a phenotype of progressive aging [40], confirming this pattern of increased “telomere attrition” previously observed in different studies conducted over HD patients with age-matched controls and relative inflammatory markers [41]. At the same time, the innate immune system is triggered by the activation of NOD-like receptors (NLRs) through several triggers such as reactive oxygen species (ROS), damage-associated molecular patterns (DAMPs), and cytokines; according to this mechanism, HD patients show upregulation of caspase-1 and IL-1β in peripheral blood mononuclear cell compared with normal controls [42].

The activation of the innate immune system is also sustained by a downregulation of regulatory mechanisms. A lack of regulation mediated by sirtuin, such as Sirtuin-1, is responsible not only for the upregulation of NF-kB [43], adding another piece to the hypothesis of a self-sustained process, but also for an imbalance toward an increase in M1 macrophage population, with an inflammatory profile, and the reduction of M2 that exerts, instead, an anti-inflammatory function [18].

In the setting of ESRD, myeloid cells are primed for the release of cytokines and ROS responding to the stimulation mediated by activators, such as phorbol 12-myristate 13-acetate or the specific TLR agonist, zymosan [44]. Clearly, innate immunity involving myeloid cells is negatively affected and the functional impairment of neutrophils and monocytes in ESRD could explain the increased susceptibility of patients with uremia to bacterial infections. Moreover, through their activation, there is a pronounced release of ROS and cytolytic enzymes, which enhances the inflammatory milieu and misleads the function of all adaptive immune cells.

At the same time, recent data suggest that mitochondrial RNA may regulate Sirtuin and one in particular, miR–125b, may be related to mitochondrial dysfunction, inflammation, and aging, also being a regulator for mTOR [45]. This evidence was confirmed by a study conducted over a 10-patient HD cohort and the genetic correlation between miRNA and uremic inflammation [46].

## 5. The Role of Complement Activation during Hemodialysis

The complement system plays a crucial role in promoting inflammation, coagulation, and oxidative burst during an HD session. Furthermore, complement activation has also been implicated in long-term chronic morbidity such as the occurrence of cardiovascular events, the development of fibrosis, and the establishment of inflammaging [26]. Complement is one of the principle components of the innate immune system, acting like a bridge of connection for adaptive response. Traditionally, complement can be activated by three pathways: the classical pathway (CP), induced by C1q binding to immune complexes or other molecules such as CRP; the lectin pathway (LP), induced by the MBL (Mannose Binding Lectin) or Ficolins recognition of carbohydrates; and the alternative pathway (AP), which is persistently low-grade activated by spontaneous C3 hydrolysis and can enhance CP and LP by C3b generation. Regardless of the pathway, complement activation will converge to cleavage of C3 and to the production of effectors components such as C3a and C3b. Increased C3b levels lead to the generation of the C5-convertase, responsible for the cleavage of C5 and the consequent production of C5a, a powerful anaphylatoxin and chemoattractant, and C5b. Next, C5b binds to the surface of the targeted biomaterial and through the interaction with C6–C9, participating in the assembly of the membrane attack complex (MAC/C5b-9) [47]. Under physiologic condition, complement activation is strictly controlled by soluble factors such as C1-INH (C1 esterase inhibitor); factor H; clusterin; and surface-bound proteins such as CD59, CR1 (complement receptor 1), and CR3 (complement receptor 3), with the latter expressed on circulating leukocytes.

During the contact of blood with hemodialysis filters, the exposed biomaterials are recognized as non-self-antigens, activating complement both in patients’ plasma (the fluid phase) and on membrane surfaces (the solid phase). This effect was firstly described by Craddock et al. [48], who originally observed an acute cardiopulmonary dysfunction in the early phase of a cellophane–membrane HD session. The complication was associated to a neutropenia and a reduction of monocytes in the first minutes of HD. Interestingly, C3 and factor B, essential AP components, were found to be significantly up-regulated.

The hypothesis that cellulosic membrane surface, by the binding of circulating C3b, could cause complement activation, with increased plasma levels of C3a and C5a during the initial phases of HD session, was investigated in 1983 by Chenoweth DE research [49]. Firstly, the main observation of the study was on the time course of complement activation: the generation of C3a reached peaks during the first 10–15 min, whereas the terminal pathway activation, with C5a and C5b-9 formation, was detectable at the end of the HD treatment. Secondly, the authors also found that hemodialysis with re-used dialyzers (i.e., dialyzers exposed to blood prior to sterilization) led to a reduced complement activation and a decreased neutropenia. From then, other investigators described subtypes of hypersensitivity reactions to the cellulose membrane defined as “first-use syndrome” as a severe, anaphylactic reaction to filter materials or to additional supplies like needles, lines, binders, plastic, and others [50,51,52,53].

Over the years, membrane biocompatibility significantly improved through from cellulose to synthetic membranes and significant progresses in sterilization methods. However, even recent synthetic filters with high biocompatibility can induce complement activation [54,55,56]. (Figure 2)

### 5.1. Alternative Pathway

In accordance with Craddock et al., other studies provided evidence on the contribution of the alternative pathway during HD treatment. In a cohort of 107 HD patients, C3d/C3-ratios, properdin, and soluble C5b-9 (sC5b-9) were found to significantly increased (up to 70%) in the plasma of HD patients at the end of treatment [57]. Notably, the measurement of plasmatic complement components (i.e., for the fluid phase C3d/C3) might not necessarily reflect the effective amount of complement activation, as it did not include the solid phase-(dialyzer bound) complement components. Increasing evidence indicated a predominant role for AP and LP involvement during HD-induced complement activation. By the proteomic analysis of dialyzer eluates, Mares J et al. provided the proof-of-concept that properdin, the unique positive AP complement regulator, could be absorbed by the HD membrane surface (solid phase), thereby promoting the AP activation in the fluid phase by the blood returning to the HD patients. (Figure 2)

Complement AP dysregulation during HD is also the result of the loss of complement inhibitors via absorption to the membrane. Yet, polysulfone membranes have been demonstrated [58] to absorb two central complement regulators: factor H and clusterin. Factor H is a crucial inhibitor of AP C3 convertase and C3b, while clusterin prevents terminal pathway activation, blocking the C5b-9 formation. Therefore, in the plasma of HD patients, the drop in the expression of these inhibitors can induce a dysregulation of the AP, leading to further complement activation in the fluid phase and to amplification of CP and LP [59]. Other studies evaluated the importance of Factor H-mediated regulation during HD, in particular, in the development of cardiovascular HD complication. Interestingly, factor H Y402H polymorphism, which affect the binding of factor H to endothelial, is an independent predictor of cardiovascular diseases in HD patients [60].

Another characterized mechanism of AP activation in HD is by the covalent binding of C3b on nucleophilic surface expressed on dialyzer membranes. These nucleophilic surfaces could be provided by cellulosic or synthetic membrane coated with albumin, IgG, LPS, and other bacterial products commonly present in dialysis solutions [61]. Controversial data are currently discussing whether the membrane adsorbed native complement can still retain the complement activation properties, as membrane binding does not necessarily imply an ability to activate fluid phase [62].

### 5.2. Lectin Pathway

In the recent years, interest has arisen in the LP activation in renal transplantation complications as well as during HD treatment. Ficolin 2 (previously named L-Ficolin), the specific pathogen recognition receptor of the LP that acts similarly to MBL, has been demonstrated to be adsorbed to polysulfone dialyzer, leading to initiation of complement cascade and to dialysis-induced leucopenia. More interestingly, a significant depletion of ficolin-2, 41% of the total, was evidenced during the HD session and correlated with C5a production at 15 min and leucopenia. Another LP protein retrieved from dialyzer eluates was the MBL [57,58]; notably, MBL depletion in the serum of HD patients was associated with a higher incidence of cardiovascular events. In addition, patients with ESKD showed a low MBL level correlating with arterial stiffness [63]. A possible explanation for the strong predictive value of MBL level for cardiovascular events could be found by the involvement of MBL in atherogenic particle removal. Consistently, a strong positive correlation between functional LP activity and C5a release was confirmed by Inoshita H. et al. [64]. The authors also observed that LP activation over time was associated with increased C5b-9 complex in the first 60 min of the HD session, remaining persistently plateaued until the end of HD.

In conclusion, these data highlighted the involvement of LP, in particular of Ficolin-2, in the C5a peaks during the early phases of HD, suggesting that Ficolin-2 adsorption to the dialyzer initiates the LP of complement activation. (Figure 2) Currently, several studies are underway to compare the level of blood proteins adsorption to dialyzer using different anticoagulants such as heparin and citrate. Heparin prevented adsorption and cleavage of several heparin-binding proteins, especially complement factor H-related protein 3, insulin-like growth factor binding proteins (2, 4, and 5), and chemerin. However, compared with heparin, citrate is associated with less protein adsorption and imperfectly crosslinked fibrin clot formation [65].

### 5.3. Classical Pathway

The classical pathway could be activated by the binding of C1q to immunoglobulin IgG adsorbed by membrane dialyzer (Figure 2). A study from Kishida suggested a role for circulating C1q for atherosclerotic cardiovascular disease in maintenance HD patients [66]. It is well known that adiponectin, an adipose-specific circulating protein with protective properties against diabetes and atherosclerotic cardiovascular disease, can bind C1q [67]. Kishida K et al. demonstrated that a low serum C1q-adiponectin/C1q ratio correlated with atherosclerotic cardiovascular disease in HD patients. Thus, besides the similarities between C1q and MBL in conformation and function in pathogen recognition and debris removal, the studies from Kishida K et al. and Poppelaars F. shed new light on discrepancy in atherogenic particles recognition and elimination during the hemodialysis. For instance, MBL is protective, whereas unbound-C1q seems to be detrimental, although the molecular mechanisms underlying these differences remain to be elucidated.

## 6. Short-Term Effect of Hemodialysis Treatment on Complement Activation

In HD, the short-term effectors functions of complement activation are the induction of inflammation, promoting coagulation, and impaired host defense owing to accelerated consumption of complement proteins [68]. Generation of C3a and C5a and opsonin (C3b, iC3b) during HD promotes inflammation by cytokines production, cytotoxic induction, and the release of granule enzymes such as elastase and mieloperoxidase (MPO) by neutrophils.

The principle consequence of complement activation in HD patients is the augmented expression of adhesion molecules, that is, the CD11b/CD18, also named CR3 on leukocytes. The activated CR3 positive leukocytes will then bind the C3b fragments on dialyzer membranes, leading to leukopenia. Importantly, the complement activated endothelium will also overexpress adhesion molecules such as P or E selectins that will induce leukocytes extravasation (and more neutropenia), together with other factors able to stimulate thrombi formation (such as von Willebrand factors). Furthermore, the interaction with leukocytes CR3 and circulating platelet will lead to thrombi formation and coagulation. The recruitment and activation of leukocytes [69] will result in the generation of oxidative stress and the release of pro-inflammatory cytokines such as interleukin (IL)-1β, IL-6, IL-8, TNF-α, monocyte chemoattractant protein-1 (MCP-1), and interferon-γ [70].

Next to regulation of oxidative stress [71], complement activation also causes the amplification of coagulation cascade, leading to the expression of tissue factor and granulocyte colony-stimulating factor in polymorphonuclear cells (PMNs), thus inducing a procoagulant state during HD [68]. This stimulus is renewed during every dialytic treatment; plasma C3 levels, before starting an HD session, were higher in patients who developed a cardiovascular (CV) event than those observed in HD patients with no CV event. Moreover, an association was found between C3 levels and the development of CV events [56].

Finally, little is known about changes in complement components on time in HD patients. C3 plasma levels decrease after 12 months compared with the baseline [56], with a negative correlation between C3 levels and the dialysis duration. In addition, the ability to activate complement has also been shown to be decreased in HD patients compared with healthy controls [72], while another study found that long-term HD patients have decreased levels of clusterin, factor B, and factor H compared with short-term HD patients [73].

## 7. Alteration in Adaptive Immune Response during Hemodialysis

### 7.1. T Cells

ESRD patients’ immune systems reveal a remarkable decrease in the lymphoid cell lineage number such as T cells, B cells, natural killer (NK) cells, and plasmacytoid dendritic cells compared with those related to the myeloid lineage such as monocytes and polymorphonuclear cells (PMNs), which might be increased in number [74]. B cells and T cells from ESRD patients have increased expression of proapoptotic molecules and are more prone to activation-induced apoptotic cell death [75,76,77]. However, this trend toward apoptosis is not sufficient to explain the decrease in number of naive T cells and the relatively normal memory T cell populations in these patients.

Thymic function in ESRD patients has been widely investigated with the evidence of a specific pattern of reduced thymic output of naive T-cells. In immunocompetent subjects, these cells express on their surface polyclonal T cell receptors αβ (TCRαβ), with a progressive circulating reduction starting from the fourth decade associated with the deposition of adipose tissue in thymus [78]. The loss of circulating naive T-cells is also associated to a loss of immunological specificity conduced to a less efficient oligoclonal TCR profile. This alteration is associated with a higher all-cause mortality rate in the elderly [79]. Malnutrition, reduced hormonal levels, reduced growth factors, and cytokines such as IL-7 or IL-2 enhance this process [80,81,82]. It is possible to hypothesize that the constant exposition to this uremic milieu might be responsible for the activation of the innate and adaptive immune system, leading to a systemic immune dysfunction. (Figure 2)

The adaptive arm of the immune system has developed in more recent times compared with the innate one, in order to offer a long-term protection from pathogens through a memory function. The partnership between B and T cells is the turning point of this system; a specific T-cell receptor is able to recognize a specific antigen bound to MHC class II molecules, normally expressed upon antigen presenting cell such as dendritic cells and macrophages. Through their TCRs, CD4 expressing T-cells, defined as helper t-cell, recognize this complex and recruit CD8 expressing t-cells with cytotoxic functions, as well as B-cells. Increased levels of p-cresol sulfate, a protein bound toxin with high molecular weight, are responsible for macrophage activation and interfere with the antigen presenting process, reducing the efficacy of this arm of the immune system [83].

The decreased numbers and impaired function of myeloid dendritic cells in ESRD patients are able to negatively affect the adaptive immune response, impairing the production of antibodies necessary for the optimal antibacterial activity of macrophages and neutrophils [84].

Effector T cell, regulatory T cells, and B cell are decreased with an inverted CD4/CD8 ratio [77]. CD4 T-cell, normally competent for antigen presentation, and CD8 T-cell, usually efficient for the cytotoxic response to viral and tumoral cells, are more prone to a state of tachyphylaxis [85,86], with an increased expression of IL-2 receptors and a severe reduction to its response. The same response involves TNF and its receptors in patients receiving RRT [86]; both CD4 and CD8 T-cells are reduced in number [87], with no further reduction determined by RRT, but just the evidence of a reduced thymic output of naïve T cell.

These particular T-cells express a greater level of CD95 that exerts a proapoptotic function and, at the same time, replicate more than they do in healthy controls, leading to a pattern of early apoptosis [88]. Moreover, regulatory functions, mediated by CD4 Treg marked with IL-2 receptor CD25+ and expressing the forehead box protein P3 (FoxP3+), are impaired and reduced in number in ESRD patients’ sera [89]. The same decline is reported for the progressive reduction of antigen-specific memory T-cells [90].

The adaptive immune system shows alteration not only in the antigen presentation phase, but also in the generation of antigen specific T and B cells. It is evident from the literature that there is a reduced thymic output of naïve T-cells, which is comparable between 40-year-old uremic patients and 80-year-old non-uremic controls. At the same time, Crepin and colleagues enlightened, according to a relative telomere length (RTL) analysis, a more sustained telomeric attrition in HD patients; RTL was comparable between 40-year-old uremic patients and 75-year-old healthy controls [84]. This condition is associated with a progressive reduction of circulating B cells that, just like T-cells, show increased apoptotic mechanisms, leading to a state of severe lymphopenia in patient with ESRD [91]. These elements contribute to a reduced tumor immunosurveillance and an increased risk of cancer.

The reduction of circulating T-cells for their impaired thymic production, associated with the inflammatory milieu that contributes to a loss of the antigen presentation function, is also responsible for a reduced humoral response to vaccination such as the Pneumococcal one [92], a low response to hepatitis B vaccination [93], and an increased risk for tuberculosis appearance [94].

Of particular interest, in recent years, new insights about the impact of CMV (cytomegalovirus) infection in HD patients and their effect on the immune system have appeared. In immunocompetent adults, according to persistent CMV infections, almost 10% of all CD4+ and more than 20% up to 50% of CD8+ T-cells are used to control the infection [95].

The Immunity in ESRD Study (iESRD) in Taiwan observed over 400 HD patients, confirming data from the literature about a lower number of circulating naïve CD4+ and CD8+ T-cells, an increased number of memory T-cells, and a more advanced differentiation of memory T-cell of uremic patients compared with healthy individuals. The increased number of memory T-cells seems to be related to HD duration [96] and higher levels of a particular PBUT, p-cresyl sulfate [97]. Interestingly, all these patients were CMV positive, confirming a 2013 paper from Meijers teamwork that firstly observed a correlation between CMV serological status and increased T-cell differentiation with premature T-cell aging in ESRD patients. Particularly, aging was related to RTL and not to a dysfunction of the telomerase enzyme. In addition, CD8+ T-cells from CMV seropositive patients presented shorter telomere when compared with age-matched CMV seronegative patients with an immunological gap of 20 years for the first when compared with the latter. No difference, instead, was observed in the setting of the CD4 cell population [98]. This difference looks like the result of a lost proliferative attitude and an increased resistance to apoptosis [99].

A 2016 publication from the same group confirmed these observations in a cohort of 49 HD patients on the waiting list for kidney transplantation when compared with age-matched healthy controls (HCs): CMV latency was a dominant factor for increased peripheral T-cells ageing in elderly ESRD patients, more than the ageing effect of uremia itself. For this reason, uremia in ESRD is believed to contribute to a decrease in anti-viral immunity, allowing more frequent CMV infections or reactivation in HD patients. The reactivation of the infection may enhance the population of CD28 null T-cells with RTL, narrowed TCR repertoire, and fewer naïve T-cells [100].

This trend toward the expansion of the CD4CD28null T-cells population is related to an increased risk of atherosclerotic diseases, promoting endothelial diseases [101,102,103]. At the same time, aging is associated with an increase of memory T-cells that re-express the CD45RA, TEMRA, and a loss of CD28, which is considered a crucial marker of immune-senescence [104]. These changes are responsible for an increased cytotoxic activity and reduced IL-2 secretion, respectively, according to TEMRA T-cells and a reduced T-cell activation, proliferation, cytokine production, and survival CD28- T-cells [105]. Interestingly, the cellular composition of the immune system does not normalize after successful kidney transplantation, despite a rapid reduction in inflammation and oxidative stress. This finding suggests that premature ageing of the immune system in patients with ESRD might be related to a permanent switch of the hematopoietic stem cell population to myeloid-generating subsets, as seen in healthy elderly individuals [1].

### 7.2. B Cells

B lymphocytes are the effector components of the adaptive immune system. After the antigen recognition by BCR (B-cell receptor), B cell (B1 subtype) can directly recognize polysaccharide and lipidic antigens and establish a response without the T lymphocyte participation. However, the response to antigen proteins required the contribution of T helper. B cells (B2 subtype) can present processed proteins by MHCII to T-helper and after activation, following the second signal by CD40-CD40L interaction and cytokine production, can differentiate into plasma cells able to secrete large amounts of antibodies.

During the HD treatment, a broad reduction in the count of B cells was observed by several studies. A possible explanation can be found by the significantly lower level of B cell activating factor (BAFF) and IL-17 receptors and in a greater susceptibility to apoptosis owing to reduced expression of Bcl-2. However, what should be clear is that not all of the B cell subpopulations are reduced in numbers. Indeed, an increase in the number of B memory cells and a decrease of immature B cells have been observed. These findings are in line with the normal antibody production observed during HD. The ratio between memory and immature cells could be explained considering that BAFF (B cell activating factor) signaling does not induce B cells’ proliferation, but enhances the survival of plasma blasts derived selectively from memory B cells. Through the BAFF receptor BR3, BAFF preferentially promotes the survival of CD38^+^ activated memory B cells and CD38^+^ rapidly dividing and activated plasma blasts. On the contrary, CD40 triggering by CD40L seems to be pivotal for B-cell growth and in particular for the proliferation of non-differentiated blasts (i.e., immature B cells) [106].

Patients undergoing HD present elevated sCD40 serum levels compared with both uremic not HD patients and healthy subjects [107], and exhibited a lack of response to HBV vaccination [108]. These data explained the low abundance of immature B cells in HD patients and could have relevant clinical implications. Interestingly, most dialysis membranes are unable to clear the sCD40; new polymethylmethacrylate (PMMA) membranes (Toray Medical Company, Japan) are emerging as therapeutic strategies to improve the humoral immune functions during hemodialysis.

## 8. Future Directions

Today, the most common HD membranes contain sulfonyl groups [109]. To further improve biocompatibility, it is vital to understand the structures that initiate complement activation as it has the potential to develop HD membranes with enhanced biocompatibility [110]. This strategy is actually followed by dialyzers manufacturers, coating philters surface not with drugs, but with particular polymers able to increase biocompatibility [111]. Particularly, the NV philter series [112] has a particular hydrophilic profile that allows an efficient coating of the dialyzer surface, thus reducing not only the anticoagulation needed during the treatment, but also platelet adhesion to the dialyzer and complement activation, also with a possible improvement of endothelial damage [113]. Vitamin-E (Vi-E) coated dialyzers are also actually available in order to reduce the oxidative burden of HD patients and improve their response to erythropoietin and endothelial stress [114]. At the same time, asymmetric philters used in the context of expanded HD (Hdx) made possible the removal of medium and large molecules with the use of solely diffusive therapies in order to avoid the drawbacks of combined diffusive/convective therapies such as albumin or immunoglobulin loss [115,116]. This particular type of dialyzers, also defined as medium cut off (MCO) membranes, is designed as high retention onset (HRO) and shows permeability close to that of a normal kidney, because it can clear proteins with a molecular weight up to 45 KD, reducing the higher albumin loss profile showed by the previous high cut off (HCO) membranes. According to their structure, MCO dialyzers extend the portfolio of removed uremic toxins, leading to a decreased general inflammatory state in HD patients [117]; are not inferior to dialyzers used for HDF (Explaining Haemodiafiltration) therapies according to the removal of λ-free light chains (FLCs), κ-FLC, β2-microglobulin, myoglobin, and IL-6; and show a comparable albumin loss profile according to a six-month observation [118]. From a clinical point of view, HDx can be successfully prescribed in the contest of cardiovascular diseases [119], secondary immunodeficiency [120], and erythropoietin resistance [121]. Of particular interest, Zickler and colleagues proved, in HDx treated patients, not only a modulation of high molecular weight toxins according to serological testing, but also a generally reduced inflammatory state as evaluated by TNF and IL-6 transcript levels in peripheral blood mononuclear cell (PBMC). According to this removal profile, MCO dialyzers could also be an efficient tool in the contest of home hemodialysis, which is actually performed only as a diffusive technique [122]. These are all clinical and biochemical features of inflammaging in the contest of HD patients.

To pursue the objective of a wider and efficient uremic toxins removal, adsorption may be a step beyond. PMMA filters are able to remove middle and large molecules such as cytokines (like IL-6); middle molecules, such as beta 2 microglobulin (B2M), removed by convective therapy, maintaining the diffusive clearance of small molecules like urea with a great biocompatibility [123]; and a lower sieving coefficient for albumin in order to avoid catabolism and malnutrition [124].

Aside from B2M, PMMA showed efficient removal of the free molecules (κ type and λ type) of the immunoglobulin light chain (Bence Jones protein). These medium weight molecules (MWM), greater than 15 kDa, accumulate at high levels in the blood of HD patients, leading to protein deposits in the internal organs acting as inhibitors of leukocyte and immune function in dialysis patients. Usually dimers (56,000 Da), these MWM are not removed by high-flux HD, but by PMMA membrane instead; according to the literature, patients affected by primary amyloidosis [125] show a reduced frequency of pain and analgesic treatment [126].

It is well known that AGEs are involved in the pathogenesis of vascular lesions, leading to an accelerated aging [127], and their rule in CKD progression is confirmed by recent literature [128]. These toxins have also been demonstrated to impair cell metabolism and trigger apoptosis, and their accumulation cannot be corrected by conventional HD or peritoneal dialysis (PD) [129,130]. Dialysis with BK-F filters is able to remove these toxic by-products, while standard HF dialyzers (with pore diameter of 30 A°) cannot.

From the literature, we also know that IL-6 regulates Th17 and Treg cell number [131], which are overexpressed in ESRD and HD patients [132,133], and which are related to major cardiovascular events [134]. PMMA proved to be efficient in IL-6 removal [135], and a 2017 paper published by Abe and colleagues evaluated the association between baseline dialyzers and all-cause mortality for one-year mortality results, suggesting that the use of different membrane types may affect mortality in HD patients, and the best performance was gained by PMMA [136].

Since the beginning, PMMA membranes were shown to be less complement activating during dialysis treatments [137]. In addition, the BK-F series proved to efficiently remove the soluble CD40, a dimeric or oligomeric protein ranging from 5 KD to 15 KD, which acts as a natural antagonist of the CD40/CD40L; the removal of this antagonist molecule is able to promote patients’ response to hepatitis B immunization [138].

The mechanism by which PMMA membranes achieve this efficacy is probably related to the absence of chemical substances capable of inducing itching such as polyvinylpyrrolidone (PVP) and bisphenol A (BPA) and the potential inhibition of cytokine production [116]. Secondly, they have protein-absorbing capacity and can absorb low and high molecular weight (HMW) proteins [139], probably pruritus-related molecules that weigh in the range of albumin or greater, which cannot penetrate sharp filters such as polysulfone membranes [140].

In 2010, Kreusser et al. [141] reported that the cumulative five-year survival rate of dialysis patients treated with the PMMA membrane is higher than that of patients treated with the polysulfone membrane (68% vs. 54%), while the 2009 Japanese Society for Dialysis Therapy (JSDT) showed better prognosis in patients treated chronically with PMMA membranes [142]. These data are consistent with a more recent publication by Abe and colleagues that, on the basis of the JSDT database, showed a lower two-year mortality risk for patients treated with PMMA and polyethersulfone (PES). Interestingly, the dataset also showed that PMMA-treated patients were older than PES-treated patients [143].

The possibility to reduce immune-senescence through dialysis has not actually been defined. Meijers and colleagues compared premature aging biomarkers in ESRD, HD-treated, and PD-treated patients [144]. Their results showed no difference between the three populations considered. Ducloux et al., instead, have recently showed a more sensible immunological profile in HD patients when compared with PD patients; both groups have similar telomere length regardless of dialysis modality, but a higher telomerase activity was evident in PD patients. Interestingly, telomerase activity was inversely related to ferritin level, while a more pronounced inflammatory state, enlightened by higher CRP and ferritin levels, was shown by HD patients [145]. This is consistent with recent evidence about the necessity to avoid, minimize, or withdraw iron supplementation in PD- and HD-treated patients because increased liver iron storage determines higher hepcidin levels with a consequent macrophage activation, which seems to be related to an increased risk of ischemic cardiovascular complications and mortality in the HD-treated population [146].

## 9. Modulation of Complement Activation during Dialysis

Therapies modulating HD-induced complement activation have focused on three treatment strategies: [1] reduction in the complement activating-capacity of the HD membrane; [2] the use of non-specific complement inhibitors (e.g., anticoagulants with a complement inhibitory property); and [3] specific complement-directed therapies.

The effect of citrate anticoagulation on complement activation has been widely studied in HD. Citrate has calcium-chelating properties, and thereby reduces complement activation [147,148]. During the initial phase of HD with cellulose membranes, citrate anticoagulation reduced C3a levels by almost 50% compared with heparin [149]. However, no complement inhibition was seen by citrate anticoagulation during HD in other studies with cellulose or synthetic membranes [150,151]. Heparinoids are also known to prevent complement activation, although this inhibition is strictly concentration-dependent [152]. Although heparin has been tested extensively in HD, none of these studies determined the effect on complement activation.

The potential of complement inhibition in HD is further underlined by the successful use of complement inhibitors for biomaterial-induced complement activation in cardiopulmonary bypass systems [153]. In patients undergoing cardiopulmonary bypass surgery, treatment with soluble CR1 (sCR1/TP30), an inhibitor of C3, leads to a decrease in mortality and morbidity as well as a reduced need for intra-aortic balloon pump support [154]. Specifically, the short half-life of sCR1 matches the need for restricted complement inhibition in HD, which is only needed during dialysis, after which complement activity should be reestablished between sessions. This approach would also prevent complications of long-term immunosuppression. Another C3-inhibitor (Compstatin), in a monkey model, has been used with a complete complement blockade during a four-hour treatment, and also for determining an increase in an anti-inflammatory cytokine like IL-10 [155]. Another therapeutic option may target early complement components such as C1; C1-INH could attenuate the activation of LP pathway [156] and also seems to be able to reduce the activation of the coagulative cascade.

Besides its central role in classic pathway complement regulation, C1-INH is involved also in the modulation of kallikrein/kinin cascades. Reduced activity of C1-INH, as observed in the genetic disorder of hereditary angioedema (HAE), results in an elevated plasma level of the vasodilator bradykinin [157]. The condition of aberrant complement activation and increased bradykinin levels in HAE resemble the hypersensitivity reactions that are often observed in HD patients. These reactions caused by anaphylactic or pseudoallergic reactions are mainly provoked by iron, erythropoietin, heparin, or angiotensin-converting enzyme inhibitors [158].

In order to reverse the acute cutaneous or abdominal attacks associated with HAE syndrome, several treatments have been investigated from the replacement therapy with intravenous C1-INH concentrates to bradykinin receptors antagonist and inhibitors of the precursor kallikrein [159].

Icatibant, a selective bradykinin B2 receptor antagonist approved by FDA in 2008, is a first-line drug for the treatment of HAE attack [160]. Several studies showed that the drug reverses increased vascular permeability in C1 esterase inhibitor-knockout mice, whereas in patients, it inhibits bradykinin-induced vasodilation and significantly improves symptoms [161]. From a dialysis perspective, certain types of dialyzers have been associated with pseudoallergic reactions. More importantly, despite the development of a recent generation of biocompatible membranes, hypersensitivity reactions are still a serious problem. A membrane commonly associated to these reactions is the high-flux polyacrylonitrile AN69 [162]. Investigations have established that the negatively charged dialysis membrane leads to activation of factor XII, which then converts pre-kallikrein to kallikrein, increasing the production of bradykinin and activating complement factors C3 and C5. These mediate the pseudoallergic reaction [163]. Although such reactions with AN69 can be seen without other predisposing factors, the use of concomitant angiotensin-converting enzyme inhibitors increases the risk significantly by decreasing the degradation of bradykinin. Therefore, the use of angiotensin-converting enzyme inhibitors should be avoided in patients who are being dialyzed using this specific membrane. Moreover, using an AN69 dialyzer coated with a biocompatible polymer that can partially neutralize the negative charge of the membrane appeared to partially alleviate these reactions [158]. Drug development efforts in HAE have also focused on target of the kinin cascade downstream of C1-INH. In the activation of the kallikrein–kinin cascade, the cleavage of high-molecular mass kininogen by the protease plasma kallikrein can generate bradykinin. Ecallantide is a 60 amino-acid recombinant protein that binds selectively to plasma kallikrein and blocks its binding site, thereby inhibiting the generation of bradykinin from high-molecular mass kininogen [164]. Ecallantide has no direct effect on the complement proteases that are dysregulated in HAE, and has a plasma half-life of ~2 hours, compared with more than 30 hours for C1-INH [158,164]. These features raised some concern about whether ecallantide would be an effective therapy in HAE. Unlike plasma-derived C1-INH concentrates, ecallantide is produced in yeast, and is thus free from any risk of viral contamination. In addition, the subcutaneous way of administration led to the opportunity that ecallantide would eventually be self-administered by patients. However, some patients who have received repeated doses of ecallantide have developed anti-drug antibodies or allergic or anaphylactic-like reactions to the drug [165]. Recently, the use of ecallantide for refractory angioedema in patients in adolescents with systemic lupus erythematosus: (SLE) with normal C1-INH levels has been investigated with promising results [166].

Finally, Lanadelumab, a human monoclonal antibody against plasma kallikrein, has been approved in several countries for the prevention of HAE attacks in patients aged ≥12 years. Lanadelumab therapy was associated with clinically meaningful improvements in HAE-specific quality of life [167]. Lanadelumab was generally well tolerated, has a low potential for immunogenicity, and offers the convenience of self-administered subcutaneous injections. In conclusion, Lanadelumab has the potential to change the approach from treatment to prophylaxis in HAE, offering a non-plasma-derived, safe, effective, and convenient prophylaxis of HAE attacks to reduce patients’ daily burden of disease and disability [168].

Eculizumab, as well as other C5a-receptor antagonists like PMX-53, may be a possible therapeutic strategy, but its permanent immunosuppressive effects may also lead to increased adverse events (e.g., infections) associated with high cost [169].

In order to reduce the problems related to the infusion of complement blockers, a possible strategy may be the use of complement inhibitors coated dialyzers; the 5C6 peptide is actually tested because of its affinity toward factor H without any effect on its function, thus providing a possible complement modulation directly on the biomaterial [170].

## 10. Conclusions

Actual data from the literature show that patients affected by ESRD have premature immunological alterations, particularly of the T-cell compartment, consistent with a pattern of immunosenescence. At the same time, inflammation and immunosenescence, combined together in the contest of inflammaging, may be considered a trigger, but also a self-sustained mechanism of this accelerated senescence, strictly correlated to the increased risk of cardiovascular events, neoplasia, and infections. RRTs also add a burden to the inflammatory milieu because of complement activation and biomaterials’ contact with blood compartment, and the lack of MMW and HMW removal is responsible for immune senescence and dysregulation. The design of new drugs, biomaterial, and devices may enhance clinical outcomes for ESRD patients on RRT and be particularly helpful for those on the waiting list, such as hyper-immune patients. More studies are needed to understand each players’ rule and, at the same time, the better therapeutic options because of their drawbacks. Finally, the knowledge of the mechanism responsible for alterations in the immune regulation in this particular setting of patients may be finally translated to the general population.

## Figures and Tables

**Figure 1 ijms-21-03422-f001:**
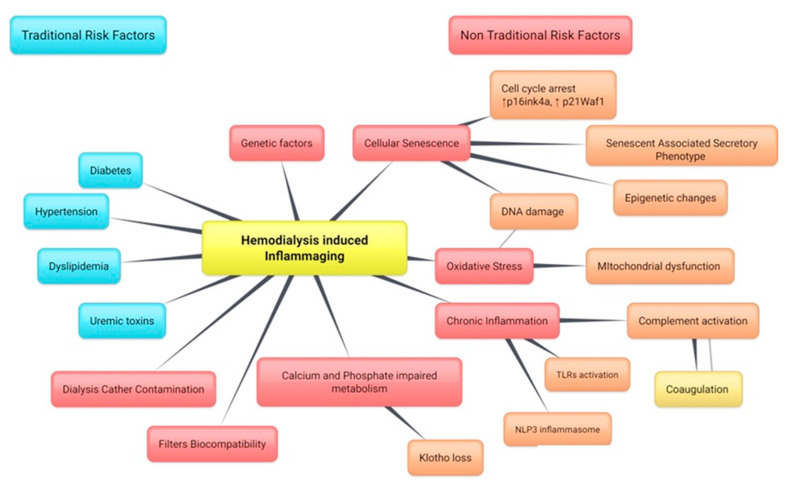
Factors involved in hemodialysis-induced inflammaging divided into traditional risk factors (in blue) and non-traditional risk factors (in red). Inflammaging is defined as the systemic, low-grade inflammation associated with increased pro-inflammatory cytokines in blood and tissues and represents a frequent cause of disability in elderly subjects. Inflammaging can be induced by a wide range of conditions such as diabetes, uremic toxins, genetic factors, or dialyzer biocompatibility. However, from the other side, inflammaging also contributes to the development and amplification of oxidative stress, cellular senescence, and persistent immune activation (i.e., complement system). The dialysis catheter contamination and the filters’ biocompatibility are exogenous risk factors that are dependent on the type of material used and the sterilization methods. On the contrary, genetic susceptibility, chronic inflammation, and the establishment of cellular senescence are examples of endogenous, patient-dependent risk factors. NRLP3, NOD-like receptor P3.

**Figure 2 ijms-21-03422-f002:**
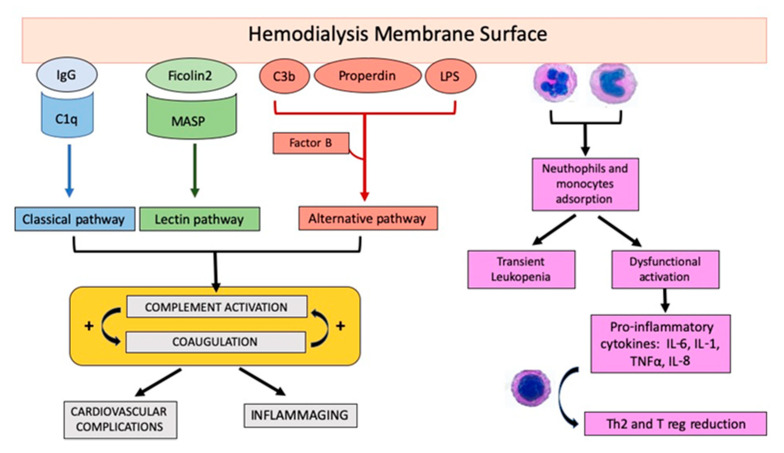
Innate immune activation on hemodialysis membrane surfaces. Hemodialysis filters can adsorb several complement components. The adhesion of circulating IgG can induce the classical pathway activation by the binding of C1q. The adsorption of Ficolin-2 to the dialyzer can lead to the lectin pathway by the binding of circulating mannose binding lectin-associated serine proteases (MASPs). Furthermore, properdin, C3b, albumin, lipopolysaccharide (LPS), or other bacterial components in dialysis solutions or hemodialysis patient bloodstream can promote the alternative route activation after the stabilization mediated by circulating factor B. The adsorption by polysulfone membranes of regulatory components as factor H, a crucial inhibitor of C3 convertase and C3b, and clusterin, able to prevent terminal pathway activation, significantly enhances the alternative pathway activation. Complement activation will result in a higher serum increase of anaphylotoxins C3a and C5a, augmented levels of soluble C5b-9, and the induction of coagulation. IL, interleukin, TNF; Tumor Necrosis Factor. Complement effectors directly boost coagulation. For example, C3a anaphylatoxin activates platelets, enhancing their aggregation and adhesion, and C5a increases blood thrombogenicity, mainly through the upregulation of TF and PAI-1 expression on neutrophils and monocytes. From the other side, the coagulation component thrombin cleaves C3 to C3a and C3b, and C5 to C5a and C5b, thus amplifying the activation of complement. The interplay between complement and coagulation system has been mainly involved in cardiovascular complication (short-term) and inflammaging and senescence processes (in the long-term). In addition, the complement system results in the recruitment and activation of neutrophils and monocytes on dialyzer membranes, leading to a transient leukopenia. The further dysfunctional activation will lead to the release of pro-inflammatory cytokines and will promote the impairment in the Th1/Th2 ratio, and thus the alterations in the subsequent T-mediated adaptive response.

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
