# Peer review of "Molecular Mechanisms of Premature Aging in Hemodialysis: The Complex Interplay between Innate and Adaptive Immune Dysfunction"

_ijms, 2020, doi:10.3390/ijms21103422_

Round 1

Reviewer 1 Report

Lossapio and coworkers analyzed clearly, in deep the numerous immune abnormalities affecting innate immunity, humoral and cellular immunities in hemodialysis patients some of which are related to uremic milieu "per se" but also related to imperfect compatibility of dialysers and devices.

I suggest that :

1° the authors analyze more deeply the work of Ducloux et al (page 13, reference 139) which shows that telomere length was inversely related to ferritin levels; in logistic regression high ferritin levels (determined by a MRI study in hemodialysis patients in the same country) predicted low telomere length. This is an important point because US authors advocated for now two decades  the use of very high IV iron to partially spare ESAs

2° The authors should have a more balanced point of view and should describe after the interest of PMMA (page 12) the interest of HDx based on the work of Zickler in Germany showing an important decrease of TNF and IL6 m-RNA in monocytes of patients.

Author Response

We thank the Reviewer 1 for the valuable and constructive comments that improved our manuscript.

We change the manuscript in accordance to these suggestions

Reviewer 2 Report

This is very striking review discussed about the possible potential mechanisms involved in inflammaging during hemodialysis. Mainly focused on innate and adaptive immune mediated mechanisms.  

I accept this review as potential publication though I see a gap in this review paper is that it does not fully address potential treatments that may be helpful. I suggest that this section might need to be expand. For example, it does not address icatibant, ecallantide, and lanadelumab. Some of these drugs are already used clinically for hereditary angioedema, a genetic disease that results in excessive complement activation.

Minor comment:

It might be easier if the sections or subjects number 6, 7 and 8 mentioned as one large section or subject. Meaning that those section 6, 7 and 8 can be mention under section 5 (Line 225) and fallowed by the 3-pathways (Alternative, Lectin and classical pathway) can explained under section 5, as 5.1, 5.2 and 5.3 instead numbering as separate sections.

Author Response

We thank the Reviewer 2 for the valuable and constructive comments that improved our manuscript.

We change the manuscript in accordance to these suggestions